# Experimental Evaluation of Collision Avoidance Techniques for Collaborative Robots

Federico Neri * , Matteo Forlini , Cecilia Scoccia , Giacomo Palmieri and Massimo Callegari

DIISM—Department of Industrial Engineering and Mathematical Sciences, Polytechnic University of Marche, 60131 Ancona, Italy
* Correspondence: federico.neri@pm.univpm.it

**Abstract:** This paper presents the implementation of an obstacle avoidance algorithm on the UR5e collaborative robot. The algorithm, previously developed and verified in simulation, allows one to modify in real time the trajectory of the manipulator with three different modalities to avoid obstacles. Some test cases with fixed or dynamic obstacles affecting the robot's motion were first simulated and then experimented on. The paper describes the hardware/software architecture of the robotic system: an external controller is realized by a standard PC that communicates with the robot controller by a TCP/IP protocol; algorithms and data processing are executed by Python/Matlab software that guarantees a duty cycle of at least 100 Hz. The error analysis between simulated and real data allows one to conclude that the developed algorithms revealed to be effectively applied to a real robotic system, showing behavior similar to what is expected by simulations.

**Keywords:** collaborative robotics; obstacle avoidance; human–robot interaction; trajectory planning



## 1. Introduction

Worker safety is one of the most important issues in the world of manufacturing today. In a decade that is projected beyond Industry 4.0, humans can work alongside robots and ensure flexible production without machine downtime. The results of this improvement are already visible in companies that have introduced new technologies and methods to reduce production costs and optimize processes. Looking to the future, we hear more and more about Industry 5.0, which envisages greater human involvement in industrial activities [1,2].

The coexistence of humans and robots is a promising goal in the industry, as it allows them to work without protective barriers, in cooperation, and with different tasks. The introduction of new technologies such as collaborative robots (cobots) makes it possible to ensure safety in this context inherently.

As described in [3], cooperation between humans and robots is very common in the industrial environment, especially in assembly lines. The main problem is that the presence of the operator near the robot opposes the speeds required for sufficient production volume. Typically in collaborative robotic cells, When an operator enters the robot's workspace, the manipulator must reduce its speed and stop immediately in case of direct contact with any part of it. This aspect may cause a slowdown in production, stopping the machine repeatedly [4,5]. In addition, robotic applications must be certified according to strict safety standards, which require the analysis of possible collisions of the robot with a human through the use of special measuring devices [6].

One way to mitigate these problems is to implement obstacle avoidance algorithms, i.e., strategies that modify the robot's motion in real-time to avoid a collision with an obstacle/human once the latter's position is sensed by a dedicated sensor system. In general, obstacle avoidance strategies are based on the combination of:

1. a method for obstacles identification and localization;
2. a control law able to modify in real time the motion of robot based on obstacles coordinates.

Numerous methods can be used to identify an object or the position of the human body in space. Studies about the prediction of human motion and the evaluation of the risk of collision can be found in [7], whereas strategies that are based on the position of the manipulator, using, for example, Newton's method for systems with high degrees of freedom, can be found in [8]. Several methods for identifying obstacles are based on contacts, in which capacitive touch sensors [9] or torque sensors [10] are used to measure the force exerted on the part of the manipulator. Other methods evaluate the position of obstacles using vision systems [11]. For example, cameras can detect the movement of the human arm (or body) by extrapolating the coordinates of skeletal models [12] and predicting their position in space thanks to machine learning algorithms [13]. In the case of depth sensors, such as RGBD cameras, it is possible to perceive the human pose in space [14,15], also by using multiple cameras around objects [16], or to detect objects in the workspace, for example through filters that isolate a specific color [17,18]. If greater precision is required, it is possible to use devices that are positioned in specific areas of the body, allowing, with an initial calibration, to identify the position of limbs in space. An example of such devices are wearable inertial measurement units (IMU) [19]. In all cases, whatever the sensor, a high acquisition frequency, at least 100 Hz, is required to ensure real-time control and capture the movement of even very dynamic obstacles.

Regarding the real-time control law, this paper resumes the obstacle avoidance algorithm proposed by the authors in [20,21]. Such a strategy allows the collaborative robot to move from one point to another with a trajectory that changes when an obstacle is perceived. When the distance between the obstacle and any point of the manipulator's kinematic chain is lower than a critical threshold, a repulsive speed is generated, which allows the robot to deviate from the path initially established, trying to avoid the collision. The region of influence of each link is defined by a radius *r* whose dimension must be set as a compromise of avoidance capability and easiness of motion and task execution. In addition to this, the region of influence may vary depending on the relative velocity between the obstacle and the manipulator. In fact, implementing a safety volume proportional to the speed of the obstacle is a technique widely used to ensure greater safety [22].

The algorithm provides three modes of motion re-planning [23]. The first mode allows both the position and orientation of the end-effector to be varied with 6-DOF. Often, however, it is necessary to limit the freedom of movement of the tool, so two other modes are available. The second mode allows 4-DOF to be varied so that all translations are free, such as rotation about the vertical axis. The third mode keeps the orientation of the tool fixed, while the position can vary with 3-DOF. The algorithm is briefly resumed in Section 2.

Section 3 describes the implementation of the algorithm on a real system with a collaborative UR5e robot that can be controlled by an external reference at a maximum frequency of 500 Hz. Several methods are available in the literature for real-time control of robots, a very common problem [24,25]. In the present application, the cobot was fixed upside down on a stand and flipped over from normal use conditions; in addition, a tool was added to make the application as real as possible. All three modes of the control law were tested.

In Section 4, the results obtained in the tests are compared with data obtained from simulations, which were carried out under the same conditions. Kinematics data of the joints are then analyzed to verify the transferability of the algorithm on a real system. To do this, 3 case studies are proposed. In the first one, the robot moves between two points in a straight line, free from obstacles. In the second test, a stationary obstacle is placed on the planned trajectory, always straight between two points, forcing the robot to avoid it by deviating from the original path. In the third example, the robot stands still at a point in the workspace while an obstacle moves to interfere with the end effector. Results and insights on future works are then discussed in the concluding Section 6.

## 2. Obstacle Avoidance Algorithm

This section invokes the obstacle avoidance algorithm in the formulation for a generic manipulator operated by a number of actuators greater or equal to 6 (depending on its degree of redundancy). The position of the joints can be described by the vector $q = \begin{bmatrix} q_1 & q_2 & \dots & q_n \end{bmatrix}^T$, in which the $n$ components give the $n$-DOF. The pose of the end-effector in the Cartesian space, according to the Euler angle ZYZ convention, is defined by the vector $x = \begin{bmatrix} x & y & z & \alpha & \beta & \gamma \end{bmatrix}^T$. The forward velocity kinematics can be written as:

$$\dot{x} = J\dot{q} = \begin{bmatrix} J_p \\ J_r \end{bmatrix} \dot{q} \tag{1}$$

where $J$ is the $6 \times n$ arm Jacobian ($n \geq 6$), with $J_p$ the submatrix of the position and $J_r$ of the rotation. To obtain inverse kinematics, it is necessary to use a damped least-square strategy to avoid singularity problems. The inverse of the Jacobian $J$ can be obtained as

$$J^* = J^T (JJ^T + \lambda^2 I)^{-1} \tag{2}$$

where $\lambda$ symbolizes the damping factor, which depends on the minimum singular value of the Jacobian matrix, and by the value $\epsilon$, which is a tunable parameter of the algorithm to which it is compared. It can be seen as a compromise between accuracy, low value, and numerical robustness, high value. In particular, for a full $J$ rank matrix, $\lambda$ is set to 0.

The algorithm detects the manipulator link where the risk of collision with obstacles is greatest. This requires calculating the distance of each obstacle from each robot body, bounded by two control points located at the extremes of the link. The center of the obstacle is called $P_o$ and its distance from the body is calculated according to the diagrams of Figure 1, in which $d_l$ is the vector representing the segment; $d_p$ and $d_d$ are the distances between the obstacle and the proximal and distal ends of the link, respectively. In addition, $P_r$ is the point of the segment closest to the obstacle. The Jacobian $J_0$ associated with the linear velocity of $P_r$ is defined as

$$J_0 = \begin{bmatrix} J_{0p} \\ J_{0r} \end{bmatrix} \tag{3}$$

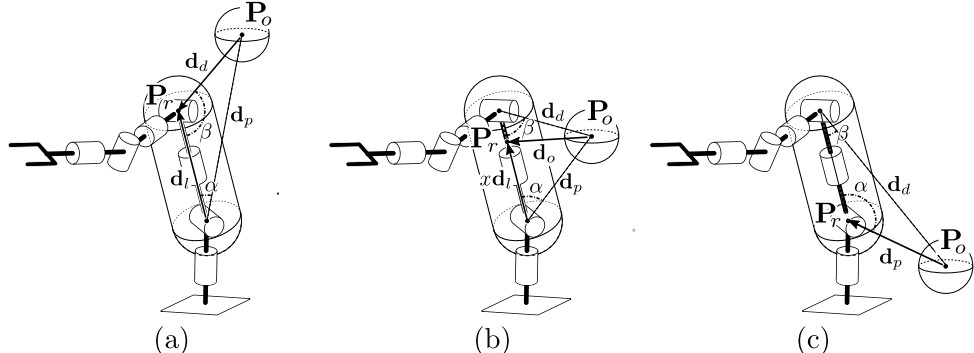

(a)  (b)  (c)

**Figure 1.** Obstacle-link distance calculation; cases (**a**–**c**) refer to Equations (4)–(6) respectively.

The motion of the robot is influenced by the presence of the obstacle if the distance $d_o$ is lower than the radius $r$, which is a parameter of the algorithm. Thus, the region of interest of each link can be visualized as a cylinder with two hemispheres at its extremities, all of radius $r$. The repulsive velocity $\dot{x}_0$ is applied to $P_r$. In detail:

(a)   $\cos \beta < 0$; the point $P_r$ is localized to the distal extremity of the link:

$$d_o = d_d \quad x = 1 \tag{4}$$

(b)   $\cos \alpha \geq 0$ and $\cos \beta \geq 0$; the distance $\boldsymbol{d}_o = \boldsymbol{P}_r - \boldsymbol{P}_o$ is orthogonal to the link, and the position of $\boldsymbol{P}_r$ along the link is defined by the scalar parameter $x$:

$$d_o = \frac{|\boldsymbol{d}_l \times \boldsymbol{d}_p|}{d_l} \quad x = \frac{d_p \cos \alpha}{d_l}, \quad 0 < x < 1 \tag{5}$$

(c)   $\cos \alpha < 0$; the point $\boldsymbol{P}_r$ is localized to the proximal extremity of the link:

$$\boldsymbol{d}_o = \boldsymbol{d}_p \quad x = 0 \tag{6}$$

Once the distance and position of the most critical obstacle are determined, three methods are proposed to avoid the collision [23].

### 2.1. Mode I: 6-DOF Perturbation

Mode I leaves the manipulator 6-DOF to avoid the obstacle, varying position and orientation with respect to the planned trajectory. The algorithm is based on the Closed-Loop Inverse Kinematics (CLIK) approach, imposing two-speed tasks. Joint velocities can be calculated as

$$\dot{\boldsymbol{q}} = \boldsymbol{J}^*(\dot{\boldsymbol{x}} + \boldsymbol{K}\boldsymbol{e}) + \boldsymbol{J}_I^* \left[ a_v v_{rep} \hat{\boldsymbol{d}}_0 \right] \tag{7}$$

where $\dot{\boldsymbol{x}}$ is the vector of planned velocities and $\dot{\boldsymbol{x}}_0 = a_v v_{rep} \hat{\boldsymbol{d}}_0$ is the repulsive velocity due to the presence of an obstacle. $v_{rep}$ is a tunable magnitude, and $a_v$ is an activation factor that is a function of the distance from the obstacle $d_0$. The damped least square algorithm is applied to invert $\boldsymbol{J}$ and $\boldsymbol{J}_I = \boldsymbol{J}_{0p}$. The term $\boldsymbol{e}$ is a vector of position and orientation errors ($\boldsymbol{e}_p$ and $\boldsymbol{e}_r$), defined as

$$\boldsymbol{e} = \begin{bmatrix} \boldsymbol{e}_p \\ \boldsymbol{e}_r \end{bmatrix} = \begin{bmatrix} \boldsymbol{P} - \boldsymbol{P}_d \\ \frac{1}{2}(\boldsymbol{i} \times \boldsymbol{i}_d + \boldsymbol{j} \times \boldsymbol{j}_d + \boldsymbol{k} \times \boldsymbol{k}_d) \end{bmatrix} \tag{8}$$

where $\boldsymbol{P}$ is the position of the end-effector and the subscript $d$ represents the desired planned variable. Instead, $\boldsymbol{i}$, $\boldsymbol{j}$, and $\boldsymbol{k}$ are the unit vectors of the end-effector reference frame. In (7), the error $\boldsymbol{e}$ is multiplied by $\boldsymbol{K}$, which is defined as $k_e \boldsymbol{I}$ with $k_e$ a gain vector that may vary according to the application. It is possible to give different gains to the orientation ($k_{er}$) or positioning ($k_{ep}$) components of the error: $\boldsymbol{k}_e = \begin{bmatrix} k_{ep} & k_{ep} & k_{ep} & k_{er} & k_{er} & k_{er} \end{bmatrix}$.

### 2.2. Mode II: 4-DOF Schoenflies Perturbation

In manipulation tasks, it is often necessary to maintain the constant orientation of the end-effector. The second mode allows only the rotation of the tool along the vertical axis. Thus, a 4-DOF motion of the Schoenflies type is used to modify the planned trajectory to avoid obstacles. Equation (7) is modified in

$$\dot{\boldsymbol{q}} = \boldsymbol{J}^*(\dot{\boldsymbol{x}} + \boldsymbol{K}\boldsymbol{e}) + \boldsymbol{J}_{II}^* \begin{bmatrix} a_v v_{rep} \hat{\boldsymbol{d}}_0 \\ \boldsymbol{0}_{2 \times 1} \end{bmatrix} \tag{9}$$

The Jacobian matrix of the mode II, of dimension $(5 \times n)$, is defined as

$$\boldsymbol{J}_{II} = \begin{bmatrix} \boldsymbol{J}_{0p} \\ \boldsymbol{J}_{r4} \\ \boldsymbol{J}_{r5} \end{bmatrix} \tag{10}$$

in which $\boldsymbol{J}_{0p}$ is the translation part of the Jacobian associated with $\boldsymbol{P}_r$, $\boldsymbol{J}_{r4}$ and $\boldsymbol{J}_{r5}$ are the first and second rows of the orientation Jacobian matrix $\boldsymbol{J}_r$ of the end-effector.

*2.3. Mode III: Perturbation with Fixed Orientation*

In the last mode, only the translation of the end-effector is allowed as a perturbation of the planned motion, whereas the orientation of the tool is constant. The inverse velocity kinematics is defined as

$$\dot{q} = J^*(\dot{x} + Ke) + J^*_{III}\begin{bmatrix} a_v v_{rep}\hat{d}_0 \\ \mathbf{0}_{3\times 1} \end{bmatrix} \qquad (11)$$

The Jacobian matrix of the second term has a $(6 \times n)$ dimension; it is composed of the translation part of $J_0$ and the orientation part of $J$:

$$J_{III} = \begin{bmatrix} J_{0p} \\ J_r \end{bmatrix} \qquad (12)$$

Thus, a null angular velocity is imposed on the perturbation term of the control law.

## 3. Implementation

This section covers the main aspects of the implementation in the real system and describes the test cases carried out to verify and validate the algorithms.

In this work, obstacles are simulated: their positions and velocities are imposed by software and communicated to the control system simulating the output of a sensor system. This is because the goal is to validate the obstacle avoidance algorithm, regardless of the methodology adopted for their detection. Second, virtual obstacles can perfectly replicate the scenarios used for simulations so that a comparison of equal conditions can be made. The integration of a real sensor system capable of detecting physical obstacles will be covered in a later article.

*3.1. System Architecture*

In addition to the coordinates of the obstacles, the algorithm needs as input the angles of the robot joints to estimate the actual relative position between the manipulator and the obstacles. Instead, the velocities of the joints to be sent to the robot controller are the output of the algorithm, and the cycle restarts by comparing the new configuration of the robot with the new position of the obstacles. Obviously, the actual operation of the algorithm depends on the speed of execution of the loop iteration; the faster the data are reprocessed and sent, the greater the responsiveness of the robot in moving away from the obstacle and, at the same time, reaching the set point. In the tests performed in this work, if no obstacles are perceived in the work area, the manipulator will follow a straight trajectory, moving the end-effector between the two established points, initial and final. This simple condition, without any perturbation, is used preliminarily to verify that the system architecture settings are properly adjusted so that the robot can move smoothly and continuously.

Figure 2 shows the hardware and software architecture of the robotic system. The control algorithm is executed by a standard PC connected to the robot controller for data exchange; a closed-loop velocity control cycle is executed iteratively until the target point is reached within a predefined tolerance.

Initially, the joint position of the robot is read using the Real-Time Data Exchange protocol (RTDE) [26], which allows synchronizing external applications with the UR controller over a standard TCP/IP connection, without affecting the performances of the UR controller. This communication method has been implemented through the Python programming language, allowing to receive data from the robot with a frequency of 500 Hz. Subsequently, this information is sent to a Matlab routine through the User Datagram Protocol (UDP), widely used in real-time applications. The Matlab routine executes the obstacle avoidance algorithm giving in the output the joint velocities reference signal to be sent to the robot controller. To do this, a TCP/IP communication is used between Matlab and a UR Script loaded on the UR5e controller that allows one to directly control the velocity of the joints of the robot by an external reference signal.

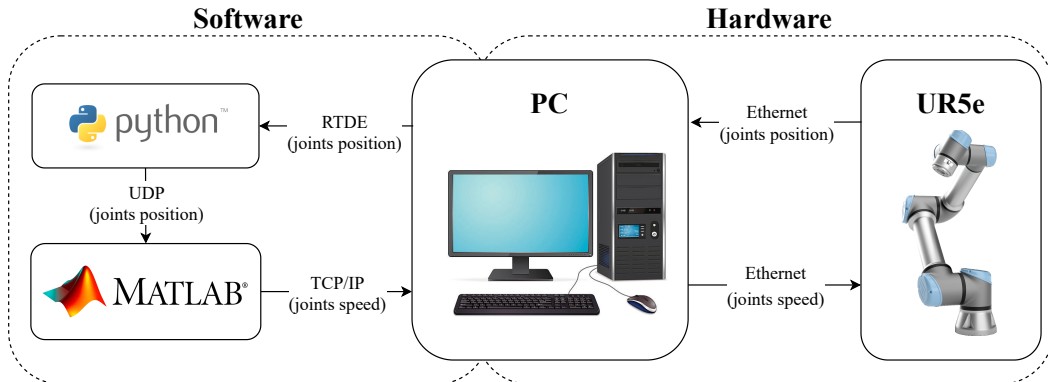

**Figure 2.** Software/hardware architecture: standard PC connected to a UR5e robot with a socket communication cable (compliant with TCP/IP standards); the PC receives in input the robot's joint rotations and sends out the joint velocities reference signal. Software is realized by Python and Matlab development environments for communication and data processing.

### 3.2. Test Cases

A particular configuration of the manipulator was chosen to carry out experimental tests representing a collaborative task between human and robot: the manipulator was fixed in reversed mounting configuration to an aluminum frame positioned at the height of 1.90 m so that the encumbrance of the base is avoided and the human can physically enter the workspace of the robot (Figure 3). In addition, a tool with a length of 205 mm has been used to simulate the encumbrance of a generic end-effector. In all test cases, a common set of parameters is used for both the simulation and the real case; in this way, it is possible to compare the results obtained since the conditions in which the algorithm is implemented are the same. Common parameters are summarized in Table 1.

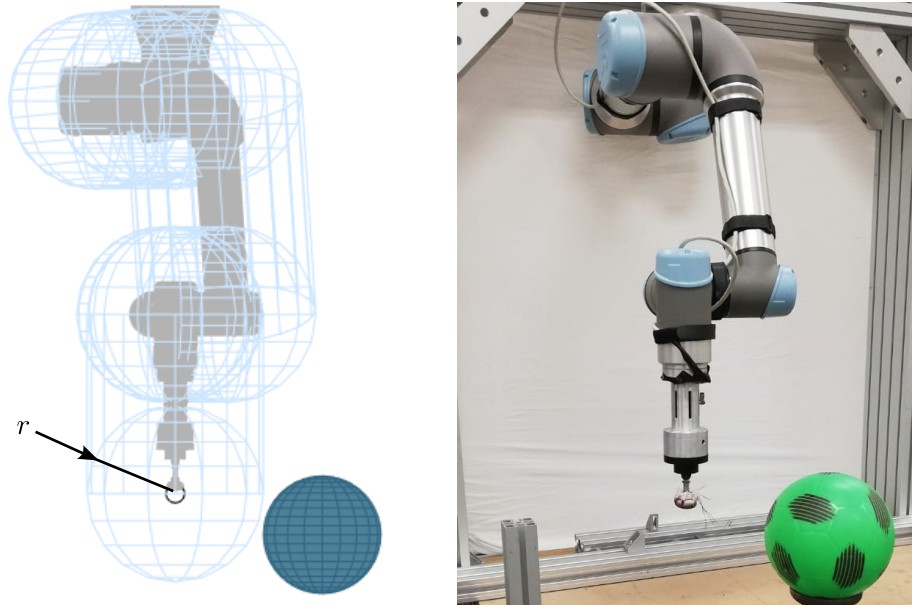

**Figure 3.** Simulated (**left**) and real (**right**) test bench.

**Table 1.** Parameter values used for both simulation and real case.

| $r_{inf}$ [m] | $r_{sup}$ [m] | $v_{inf}$ [m/s] | $v_{sup}$ [m/s] |
|---|---|---|---|
| 0.15 | 0.20 | 0.1 | 0.5 |
| $\lambda_{max}$ | $\epsilon$ | $\dot{\theta}_{max}$ [rad/s] | $T$ [s] |
| $10^{-3}$ | $10^{-3}$ | $\pi$ | 5 |

As mentioned, the position of the obstacle is simulated to avoid possible errors in the object detection system. In the proposed examples, a general spherical obstacle is considered. Regarding the safety volume around the links of the robot, the radius $r$ is set as a linear function of the velocity $v$ of the obstacle, with lower and upper bounds ($r = r_{inf}$ for $v \leq v_{inf}$, $r = r_{sup}$ for $v \geq v_{inf}$). The $\dot{\theta}_{max}$ value is introduced to saturate the angular velocity of the joints if the algorithm generates a speed excessively high. This safety limit is imposed at all joints, and the consequent positioning error is recovered in subsequent movements thanks to the proportional corrective term.

Other parameters must be changed in the real system because the simulation operates in an ideal context that does not consider the actual reaction of the robot's motors, the communication latency, or other implementation and hardware limits. Gains parameters, in particular, such as the repulsive speed $v_{rep}$ and the proportional gain $k_e$ of Equation (7), resulted very sensible to variations and have been tuned by a trial and error procedure. Final values are collected and discussed in the next section.

## 4. Results

Three different test cases were performed to verify the algorithm's applicability to a real system. For each case, four frames ranging from $T = 0$ to $T = 5\,\text{s}$ are plotted to show the robot's motion. In addition, the position and the velocity of each joint and the respective absolute errors over time are plotted. For Sections 4.2 and 4.3, the 3 modes of obstacle avoidance are tested.

### 4.1. Test Case 1

In the first test case, the robot moves between two points in a straight line without obstacles. This case is used to verify the communication system and to generate a control case for other cases with obstacles. Figure 4 shows the frames at $t = [0; 2; 3; 5]$ s for both the simulation and the real case.

In the plots shown in Figure 5, the curves of the simulation and the real case perfectly overlap, showing that the robot can follow the given path and reach the final position in the expected time. The absolute error between the two curves (Figure 6) is extremely small and shows a constant trend, with an average of about 0.0025 rad for the position and 0.003 rad/s for the velocity. Regarding the communication between the PC and robot controller, the transmission reached a rate of about 350 Hz, which is higher than the desired minimum value. Compared to the simulation, there is a slowdown due to data exchange and processing; nevertheless, the error generated is negligible, and the robot moves smoothly without sudden changes in velocity.

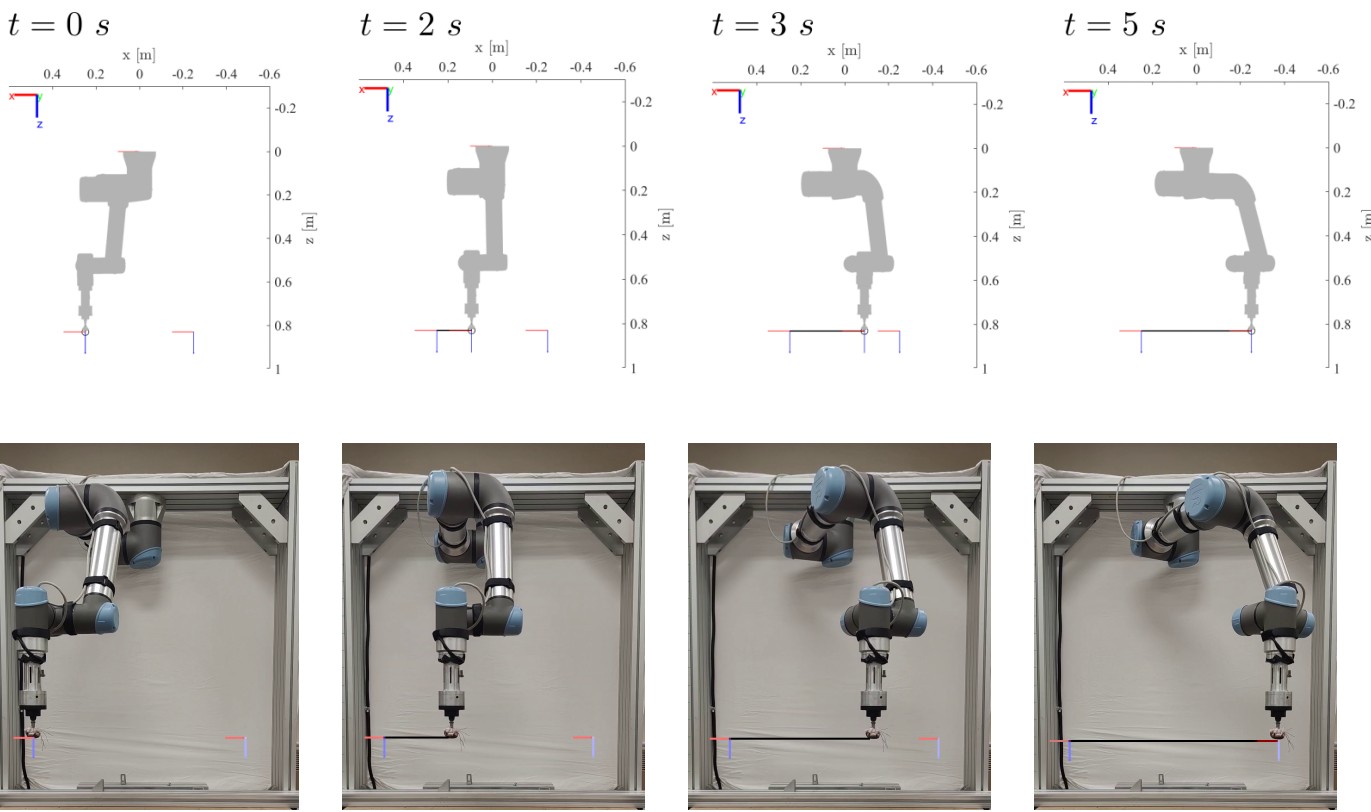

**Figure 4.** Test case 1: straight motion between two points, Mode I. Simulated (**top**) and real (**bottom**) motion of the robot.

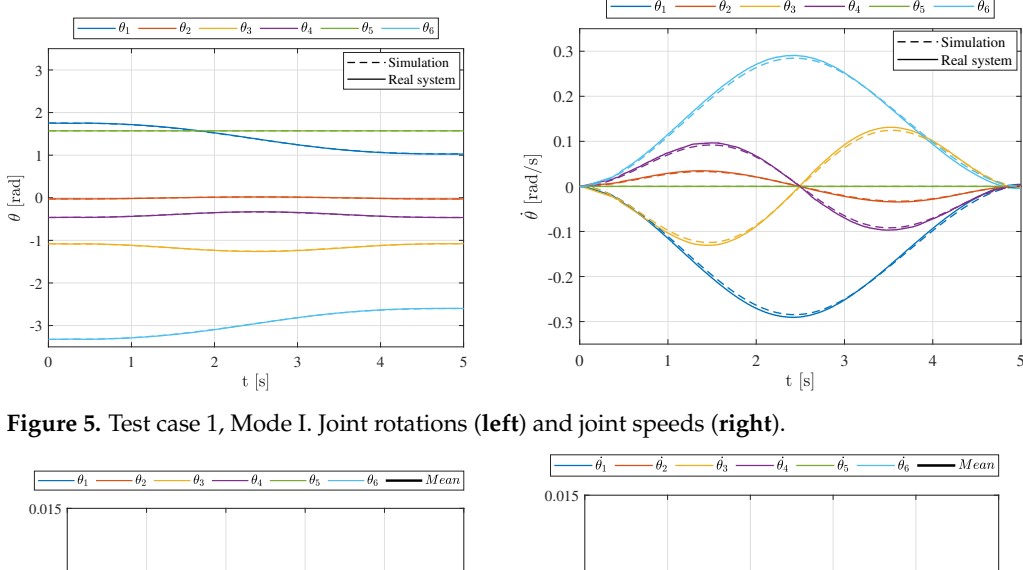

**Figure 5.** Test case 1, Mode I. Joint rotations (**left**) and joint speeds (**right**).

**Figure 6.** Test case 1, Mode I. Error between simulation and real system. Joint rotations (**left**) and joint speeds (**right**).

### 4.2. Test Case 2

In the second test case, a stationary obstacle is placed in the robot's path. As in the previous example, the planned motion is a straight line between two points at a speed of 0.12 m/s. A spherical object has been placed at the location where the virtual obstacle is located to facilitate the interpretation of the motion (Figure 7). When the end effector's region of influence reaches the obstacle, the algorithm imposes the repulsive velocity term. Once the robot overcomes the obstacle, the positioning error is gradually recovered, thanks to the proportional corrective term.

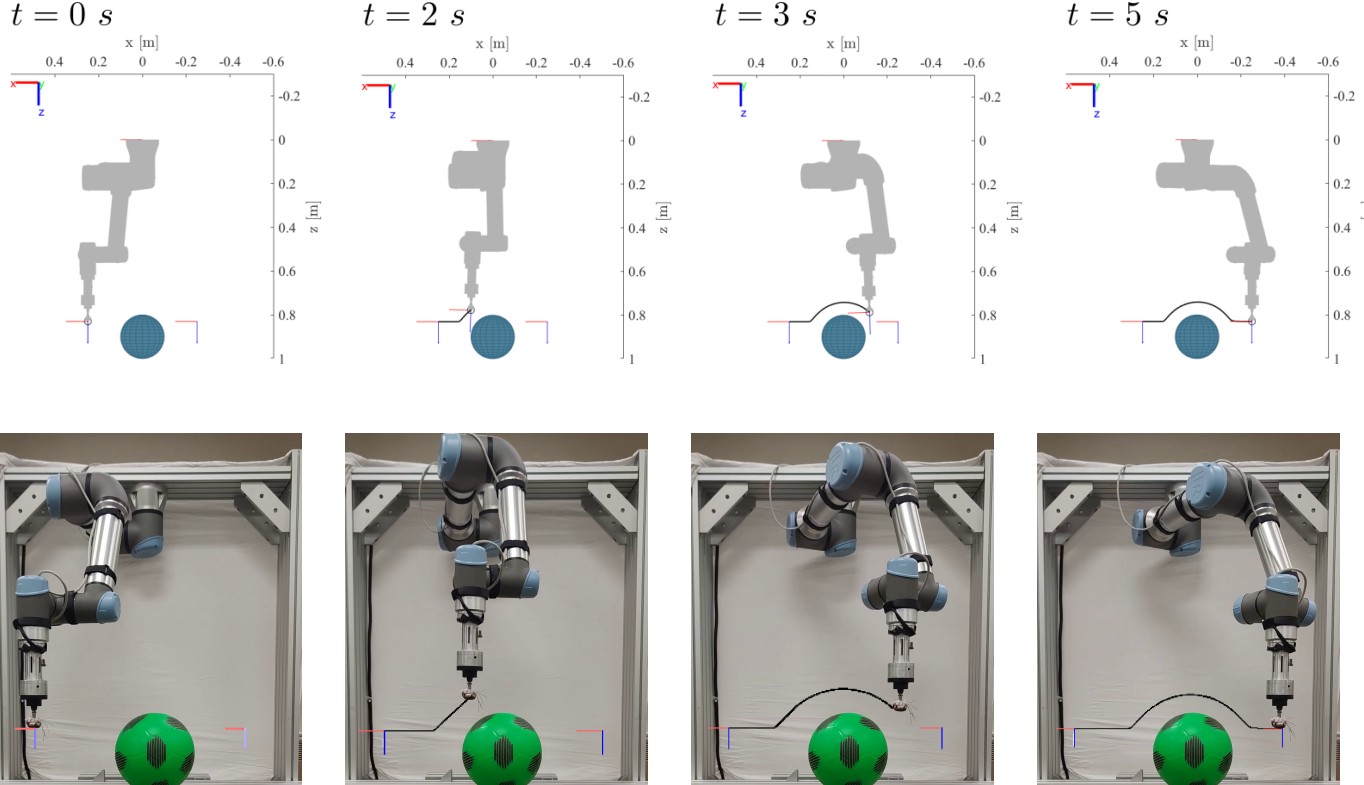

**Figure 7.** Test case 2: avoidance of a fixed obstacle while the robot moves along a linear path, Mode I. Simulated (**top**) and real (**bottom**) motion of the robot.

Figures 8–10 make a comparison between the joint positions and velocities in the simulation and in the real system. Figure 8 refers to the case where the robot avoids the obstacle with mode I, i.e., the end-effector can freely vary both the position and the orientation. For this reason, a change in the velocity of joints 4 and 5 can be observed at $t = 2.2$ s. Figure 9 refers to mode II, i.e., with the possibility of changing the position and only the rotation about the z-axis. It can be seen that there is a small variation in the velocities of joints 4 and 5 so that the obstacle can be avoided without changing the orientation of the end-effector. Figure 10 refers to mode III, where the orientation is kept constant. The diagrams are very similar to the previous case, except for a different tendency of the speed of joint 6.

All plots show that the simulated and real data follow the same trend. The main difference is due to a slight latency in the real system when the obstacle effect is activated, which leads to an increase in joint velocity so that the obstacle can be avoided in a shorter time. Nevertheless, the recovery in the final phase of the trajectory is fast enough to reach the endpoint in the foreseen time.

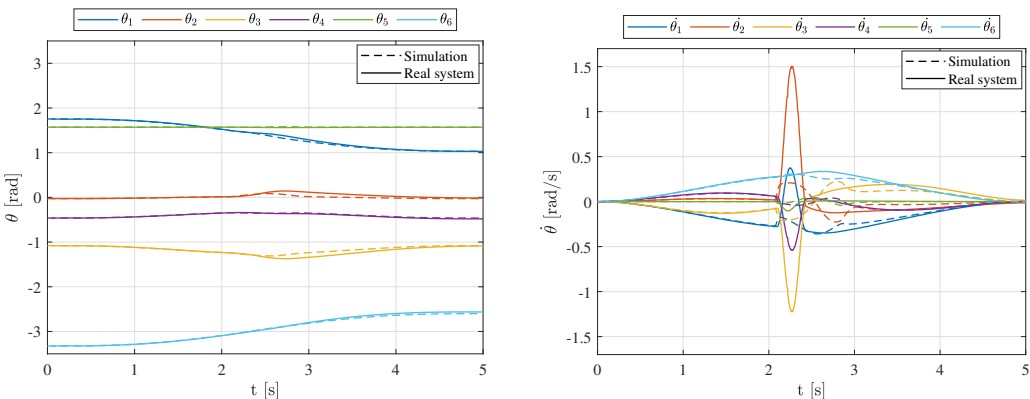

**Figure 8.** Test case 2, Mode I. Joint rotations(**left**) and speeds (**right**).

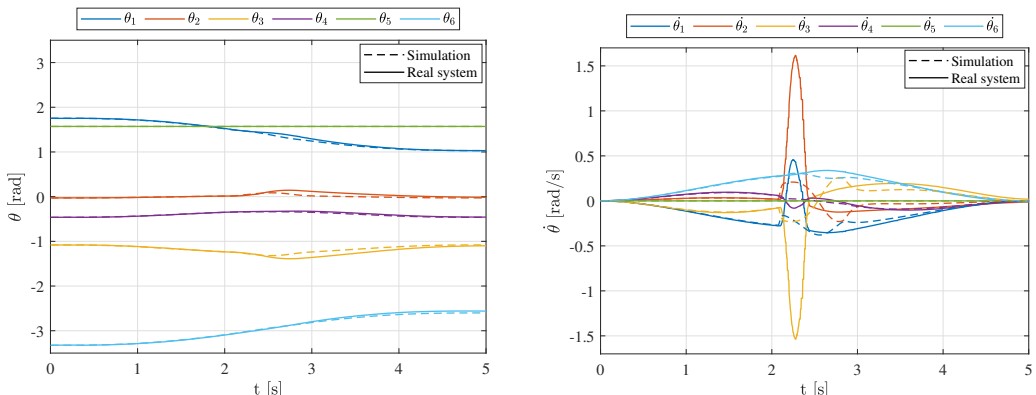

**Figure 9.** Test case 2, Mode II. Joint rotations (**left**) and speeds (**right**).

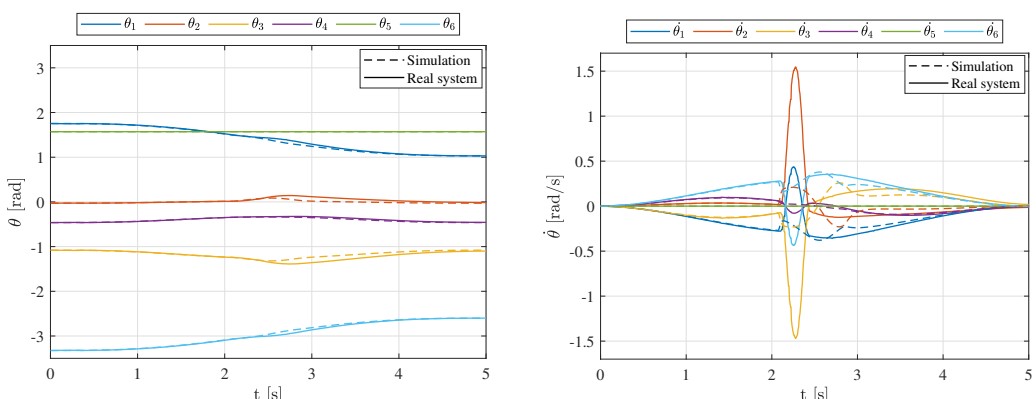

**Figure 10.** Test case 2, Mode III. Joint rotations (**left**) and speeds (**right**).

Figures 11–13 show the error between simulated and real data when mode I, II, and III are selected, respectively. A peak of the error is visible when the obstacle is perceived, but it does not exceed 0.14 rad in position. The average of the error between the joints is 0.02 rad for position and 0.1 rad/s for velocity.

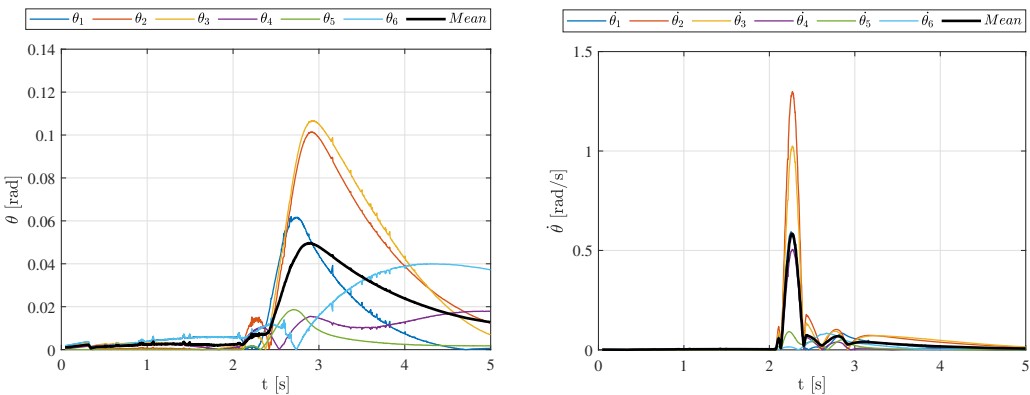

**Figure 11.** Test case 2, Mode I. The error between simulation and real system. Joint rotations (**left**) and joint speeds (**right**).

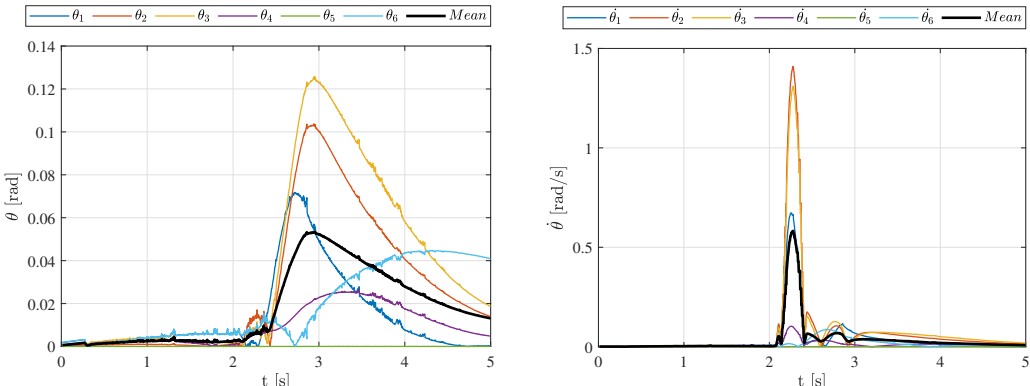

**Figure 12.** Test case 2, Mode II. The error between simulation and real system. Joint rotations (**left**) and joint speeds (**right**).

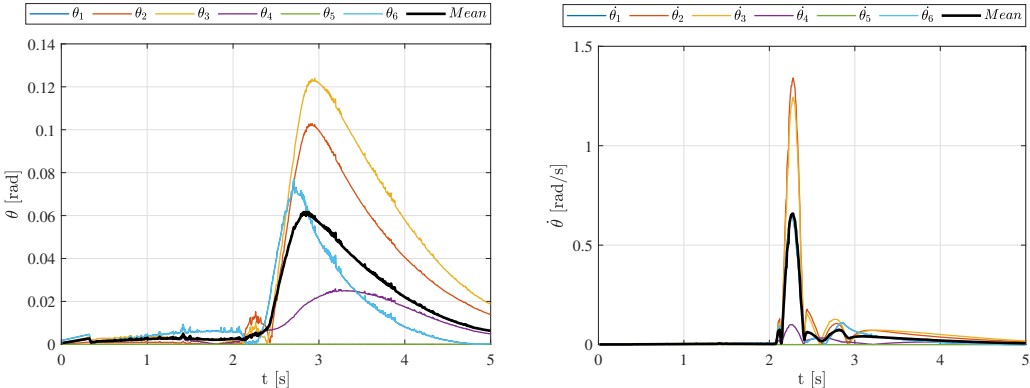

**Figure 13.** Test case 2, Mode III. The error between simulation and real system. Joint rotations (**left**) and joint speeds (**right**).

### 4.3. Test Case 3

In the last case, the obstacle moves on a straight line between two points while the robot takes a fixed position in the middle of the trajectory (Figure 14). The obstacle moves at a speed of 0.11 m/s, higher than the reference speed $v_{inf}$; thus, the safety radius $r$ results increased than $r_{inf}$. When the obstacle reaches the end-effector's region of influence, the algorithm demands a repulsive velocity from the robot that starts to move. Once the obstacle leaves the region of influence, the positioning error is recovered, and the robot returns to its stationary pose.

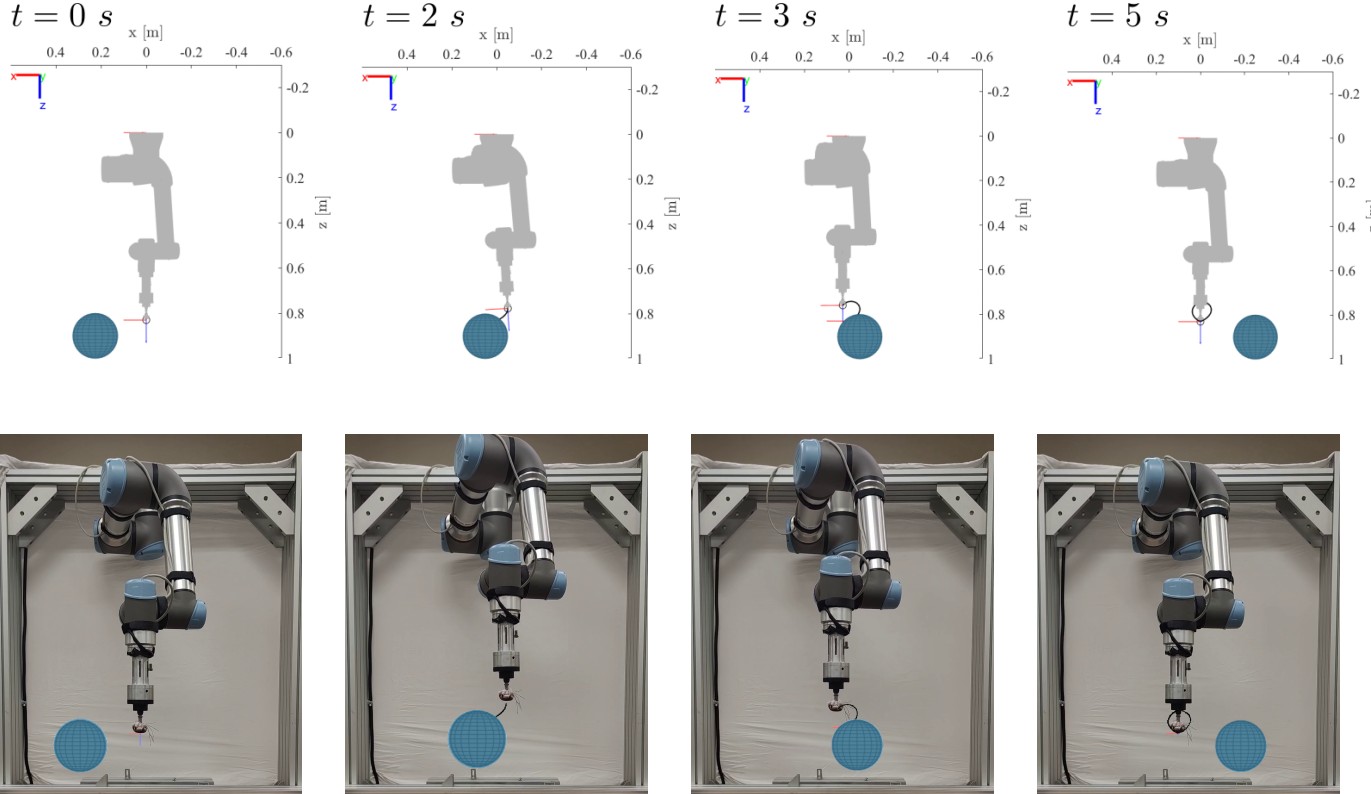

**Figure 14.** Test case 3: avoidance of a dynamic obstacle interfering with the end-effector in a fixed position, Mode I. Simulated (**top**) and real (**bottom**) motion of the robot.

Figures 15–17 show the graphs of joint positions and velocities in the three avoidance modes. Similar considerations to case 2 can be made by analyzing the different responses for the various modes.

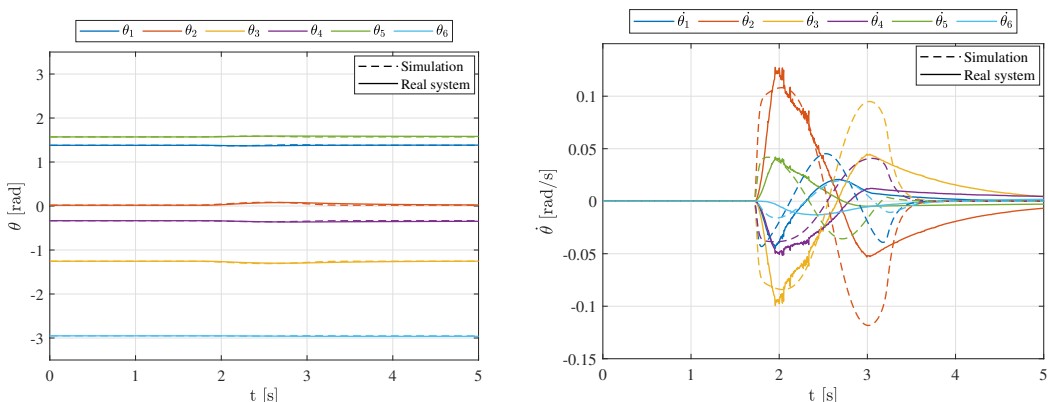

**Figure 15.** Test case 3, Mode I. Joint rotations (**left**) and speeds (**right**).

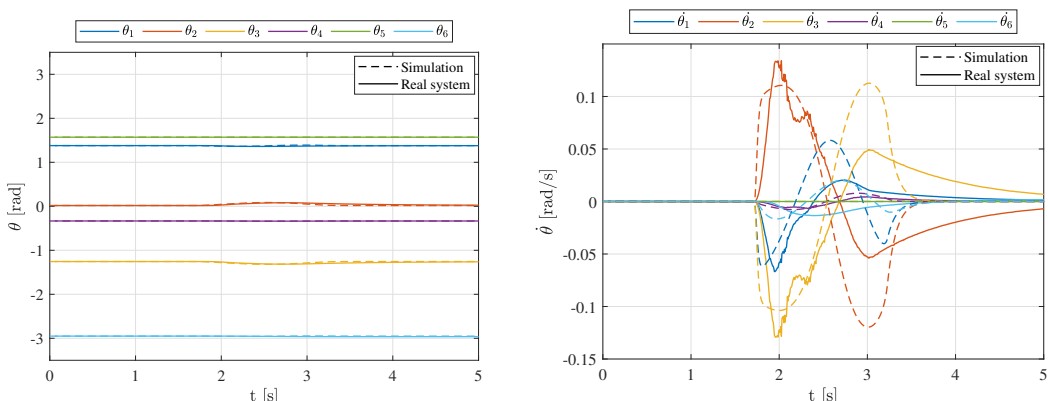

**Figure 16.** Test case 3, Mode II. Joint rotations (**left**) and speeds (**right**).

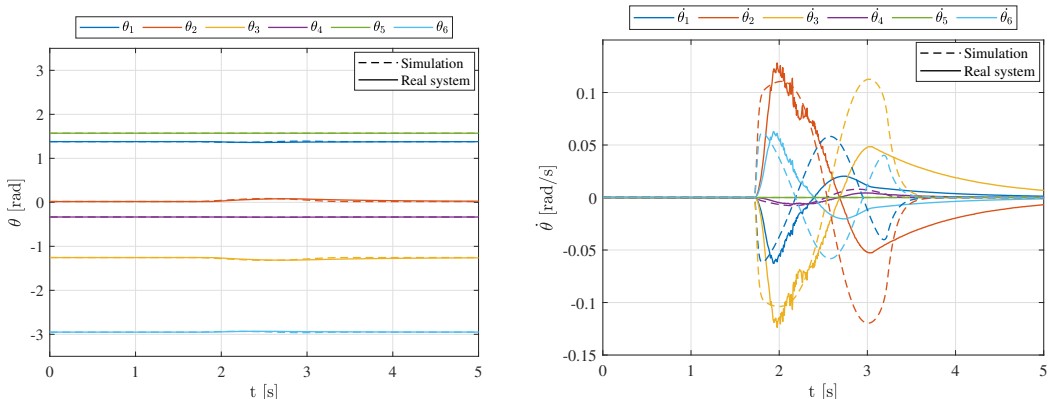

**Figure 17.** Test case 3, Mode III. Joint rotations (**left**) and speeds (**right**).

Figures 18–20 show the trend of the error between simulations and real data. The absolute error increases, as in the second test case, from the moment the obstacle is perceived but remains below 0.04 rad in position. The average of the error is about 0.005 rad for position and 0.01 rad/s for speed. As a summary of the results obtained by comparison between simulated and experimental data, Table 2 reports maximum and average errors of joint positions and speeds for all the examined cases.

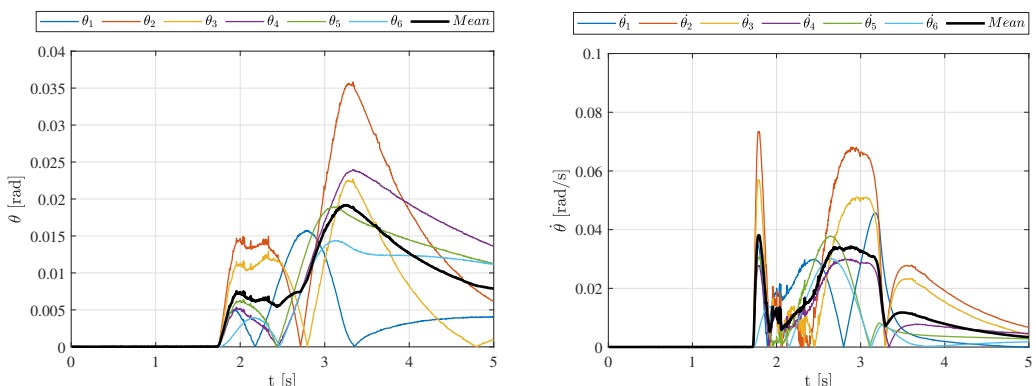

**Figure 18.** Test case 3, Mode I. The error between simulation and real system. Joint rotations (**left**) and joint speeds (**right**).

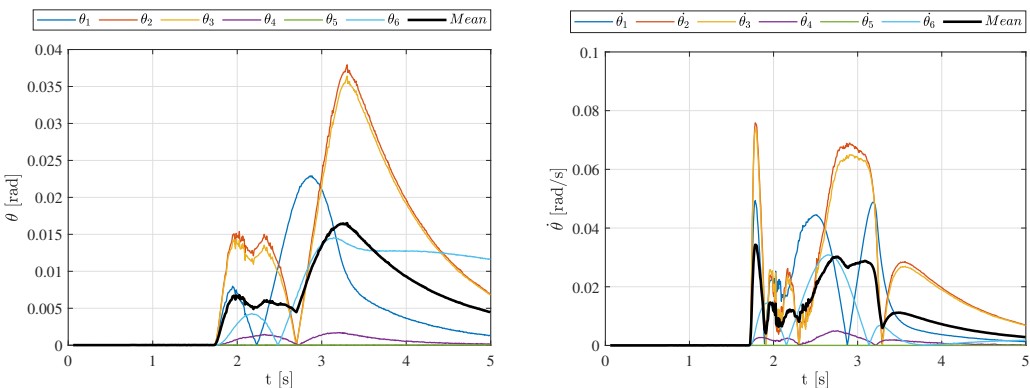

**Figure 19.** Test case 3, Mode II. The error between simulation and real system. Joint rotations (**left**) and joint speeds (**right**).

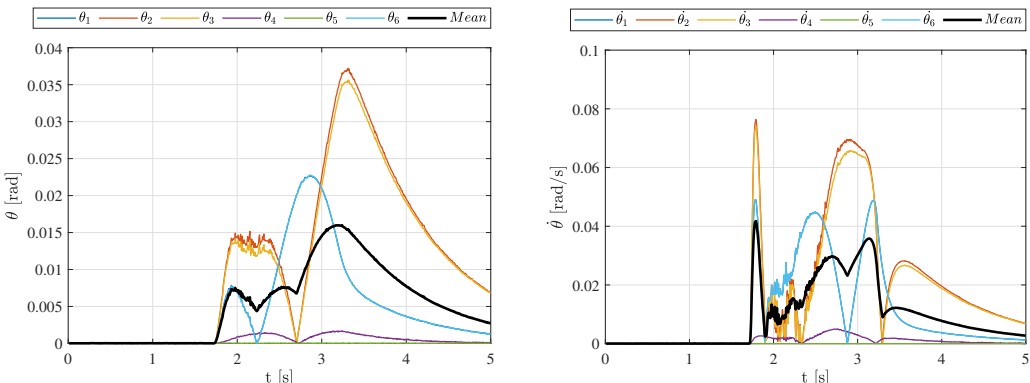

**Figure 20.** Test case 3, Mode III. The error between simulation and real system. Joint rotations (**left**) and joint speeds (**right**).

**Table 2.** Summary of results: comparison between simulated and real joint data.

|  | Max Position Error [rad] | Max Speed Error [rad/s] | Average Position Error [rad] | Average Speed Error [rad/s] |
|---|---|---|---|---|
| Test case 1 | 0.008 | 0.008 | 0.003 | 0.003 |
| Test case 2 | 0.13 | 1.50 | 0.02 | 0.10 |
| Test case 3 | 0.04 | 0.08 | 0.005 | 0.01 |

The tests described in this chapter were performed after carefully evaluating the coefficients that allow the algorithm to perform optimally. Some of these values, particularly the proportional gains, cannot be determined a priori. These gains were modified by trial and error until a good fit was obtained between simulated and real data; it was also verified that the robot motion was continuous and smooth. However, it should be noted that the absolute error between the simulations and the real case would have been acceptable even with the same parameters used in the simulation. In detail, the values for $v_{rep}$ and $k_e$ used in simulation and the real system are listed in Table 3.

**Table 3.** Values of the algorithm parameters for the simulation and for the real case.

|  | $v_{rep}$ [m/s] | $k_{ep}$ [s$^{-1}$] | $k_{er}$ [s$^{-1}$] |
|---|---|---|---|
| Simulated | 2 | 10 | 30 |
| Real | 1 | 1 | 0.16 |

## 5. Discussion

In all the avoidance modes, the differences between simulated and real data resulted less than 0.02 rad for joint positions and 0.1 rad/s for joint speeds. Thus, it can be concluded that the theoretical algorithms revealed to be effectively applied to a real robotic system showing behavior similar to what simulations expect. Only a few parameters, i.e., proportional gains, need a trial and error setting procedure to optimize the response of the system.

Diagrams of the three modes in which the robot can execute a trajectory to avoid the obstacle were proposed for each test case. In this way, it is possible to analyze the behavior of the system in the case when some degrees of freedom related to orientation cannot be updated during motion. Even though only the three translation-related DOFs are exploited in Mode III to replan the motion, the system still shows good obstacle avoidance capability, and the variation in position and velocity between the real and simulated data remains negligible: the maximum position error in mode III is about 0.13 rad, whereas the same value for mode I is 0.11 rad. In general, the difference between simulated and real data may depend on many variables. The most important ones are

1.  the communication protocol between the external controller and the robot, thus, the frequency of data exchange;
2.  the accuracy of the robot controller in driving the motors via internal control loops to follow the external speed reference signal; this characteristic may vary among different manufacturers or robot models;
3.  the acquisition frequency of external sensors for obstacle detection, which should be as high as possible or at least comparable to the control frequency;
4.  the complexity of the algorithm, which, if too high, can increase the computation time, thus, reducing the speed of the control loop.

In this work, the use of a UR5e robot allowed easy integration with the external controller at a frequency of about 350 Hz, a remarkable achievement when compared with previous work (e.g., 250 Hz in [27]). However, so far, the position of the obstacle has been generated virtually to eliminate any sensor error. Future integration of a real sensor system could lower the control loop frequency, which should still remain high enough to ensure smooth control of the robot, as experienced in [28] where a final control rate of about 70 Hz revealed sufficient to have an efficient real-time control. In summary, the results show that the algorithm is indeed implementable on the real system and processable even by a standard PC, thus, having a low economic impact.

## 6. Conclusions

In this paper, an obstacle avoidance strategy for manipulators moving in dynamically changing environments is implemented and tested. Starting from a theoretical framework developed by the authors, a control and communication system is introduced to implement the algorithm on a UR robot. Three test cases have been proposed, which have been analyzed to verify the operation of the algorithm and compare the real behavior of the robot with the simulations used to develop the algorithms. In the present work, obstacles were generated virtually by software to decouple errors due to the control law from those due to an external obstacle detection system. Now that the algorithms have been validated, the next step in the study will be implementing a real sensor system, or different systems, for obstacle detection. Further future work will envisage the implementation of mobile manipulators consisting of a collaborative 7-DOF robot positioned on an omnidirectional mobile base, using algorithms already simulated in previous works [29]. In this way, the additional degrees of freedom of the system can be exploited to improve collision avoidance and confer the robot an optimal motion in terms of smoothness and dexterity.

**Author Contributions:** Conceptualization, G.P. and C.S.; validation, G.P.; investigation, F.N. and M.F.; writing—original draft preparation, F.N.; writing—review and editing, G.P.; supervision, G.P. and M.C. All authors have read and agreed to the published version of the manuscript.

**Funding:** This research received no external funding.

**Conflicts of Interest:** The authors declare no conflict of interest.

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
