# Peer review of "Experimental Evaluation of Collision Avoidance Techniques for Collaborative Robots"

_applsci, doi:10.3390/app13052944_

Round 1
Reviewer 1 Report
-In the introduction, the basic information about the subject and the studies in the literature on this subject are adequately explained.
- In the article, real-time avoidance algorithms for obstacles are proposed, but how obstacles are detected is not explained. Entering the obstacle coordinate as a parameter indicates that the proposed algorithm is not real-time. Authors need to clarify this issue.
- The proposed algorithm has been tested with 3 different methods both in the simulation environment and experimentally and the results are discussed.
- The findings obtained should be compared with the results of similar studies in the literature.
Reviewer 2 Report
This manuscript presents the implementation of an obstacle avoidance algorithm on the UR5e collaborative robot. The algorithms and data processing are executed by Python/MATLAB software. The error analysis is executed. On the whole, the research of this manuscript has certain significance, but there are many problems that need further revision.
1. In the manuscript, the variation in the formula should be italic, and the vector and matrix should be italic and bold. Please revise all the formulas in the manuscript.
2. In Section 4, it is suggested to give a table that comprehensively lists Case 1 to Case 3, so that readers can comprehensively understand and distinguish the differences between Case 1 and Case 3.
3. In line 222-line 223, Whether “For cases 4.2 and 4.3, the 3 modes of obstacle avoidance are tested.” should be changed to “For cases 4.1 and 4.3, the 3 modes of obstacle avoidance are tested.”
4. In addition to the color difference, all the data curves in the manuscript suggest that the line characteristics change. If the reader prints this manuscript, there is no way to judge the angles and angular velocities of joint 1 to joint 6.
5. In Figure 14, the real scene given by the authors, such as a sphere, should be a real body rather than a virtual body. It is suggested that the author really give the experimental scene to increase the persuasiveness of the manuscript.
6. In this manuscript, the authors give Case 1: the robot moves between two points in a straight line without obstacles, Case 2: a stationary obstacle is placed in the robot’s path, and Case 3: the obstacle moves on a straight line between two points. Then, the authors are suggested to give Case 4: the object moves on a curve line.
7. In the writing rules of the manuscript, the statement "... is as shown in the Figure/Table X" should be given first, and then the corresponding figure/table should be given. On the contrary, it is not allowed. Hence, the position of Figure 14 should be modified. It is suggested that the authors check whether there are similar problems in the full manuscript.
8. In the Abstract, the authors say that the algorithms and data processing are executed by Python/MATLAB software. Hence, in order to increase the integrity of the manuscript, the algorithms and data processing should be given by Python/MATLAB software.
Reviewer 3 Report
A very interesting research work, with a well formulated research approach and validation. The title and abstract are appropriately formulated and the research article follows this logic and argumentation. A few minor notes for improving the article are provided hereunder:
- A few grammatical mistakes need to be addressed. Some of these have been highlighted in the attached document.
- A comparison to other state of the art approaches to collision detection is found missing. Especially a comparison to the work of Lihui Wang, such as his article on "Active collision avoidance for human–robot collaboration driven by vision sensors"
- Figure 7, images (b) and (c) do not look to be sequential. Can this be addressed or explained? (see highlighted zone in the attached document)
- In the discussion it would be interesting to discuss how this would be effected if both the robot and the object (human) are in movement. This particular case study is not evaluated, and it would be important to know why not, and what are the expected results of this.
- In figure 2, it is not clear why only joint speed is being sent to the robot controller. If this is a closed loop feedback system, then it should be mentioned within the text, because I believe that this is not clearly discussed. This could easily be addressed in line 181: a closed loop velocity control cycle is executed iteratively until the target point is reached within a predefined tolerance.
Reviewer 4 Report
The topic discussed in the article is relevant and important. This paper presents the implementation of an obstacle avoidance algorithm on collaborative robot. The algorithm allows to modify in real time the trajectory of the manipulator with three different modalities in order to avoid obstacles.
The Introduction is sufficiently detailed, and a number of sources are reviewed. Most of the sources are from recent years and reflect the current situation in the subject matter. The Second Section deals with the obstacle avoidance algorithm. The hardware/software architecture of the robotic system is described. The Third Section covers the main aspects of the implementation in the real system and describes the test cases carried out to verify and validate the algorithms. The results of the research are presented in the Fourth Section. The presented conclusions are based on the results of the conducted research.
Overall, the material in the article is presented in a coherent and logical manner. However, in order to make the article clearer for the readers, it would be good to check that all the notations used in the tables and figures are properly explained. The information shown in the figures is not clearly presented, it is quite difficult to understand the essence just from the names of the pictures and the presented images.
Quite a lot of research material and results have been presented, so it would be logical to precede the conclusions with a discussion of the results and their comparison with the results obtained by other scientists.
Reviewer 5 Report
The paper presents an obstacle avoidance algorithm for robotic manipulators, evaluated both analytically and experimentally. The quality of the manuscript is good, and I suggest that it can be considered for publication after the authors have addressed a few minor points:
1. It appears that Figures 1 (a) and 1 (b) are switched with respect to Equations (4) and (5), please check.
2. It is not clear from Section 2 why the algorithm has to sort the position of the obstacle into the three categories described by Eqs. (4-6). How are the three cases treated differently?
3. What is the advantage of implementing part of the algorithm in Phyton and part in Matlab? Is there an improvement in performance over using a single programming platform? or has it to do with the availability of libraries?
4. The authors point out that some gains affecting the behavior of the algorithm have to be tuned manually by trial and errror, especially for the physical robot. Can the setting of those gains be automated?
Round 2
Reviewer 1 Report
Thanks for the edits and explanations.
Author Response
Thanks
Reviewer 2 Report
The author of this manuscript has not made effective amendments to the manuscript according to the reviewer's suggestions, and the reply to the reviewer's suggestions is very perfunctory. The overall academic quality of this manuscript is not high, and some formats are not implemented according to the requirements of the journal. It is suggested that the author download several articles that have been published in Applied Sciences and take a good look at their formats. The reviewer's opinion is to help the author improve the overall quality of the manuscript. The reviewer and the author are friends rather than enemies. As a scientific researcher, you need to keep a humble attitude, accept other people's suggestions, examine the problems of your manuscript, and strive to improve the quality of your manuscript. The reviewer put forward eight questions, and effectively revised the manuscript based on the second question.
Reviewer 4 Report
Thanks to the Authors for the corrections!
I understand that in this case the comparison with the research of other scientists is not the main purpose of this article. But in my opinion, a scientific article should have a general discussion of the results at the end, where the scientific novelty of the research results should be highlighted (preferably in comparison with other similar studies). In this case, excluding descriptions of individual test cases, there is practically no such discussion at the end of the article.
